# Bacillus Calmette–Guérin Vaccine Attenuates Haloperidol-Induced TD-like Behavioral and Neurochemical Alteration in Experimental Rats

**DOI:** 10.3390/biom13111667

**Published:** 2023-11-19

**Authors:** Narhari Gangaram Yedke, Shubham Upadhayay, Randhir Singh, Sumit Jamwal, Sheikh F. Ahmad, Puneet Kumar

**Affiliations:** 1Department of Pharmaceutical Sciences and Technology, Maharaja Ranjit Singh Punjab Technical University, Bathinda 151001, India; yedkeng16a@gmail.com; 2Department of Pharmacology, Central University of Punjab, Ghudda, Bathinda 151401, India; 3Department of Molecular Psychiatry, Yale University School of Medicine, New Haven, CT 06510, USA; 4Department of Pharmacology and Toxicology, College of Pharmacy, King Saud University, Riyadh 11451, Saudi Arabia

**Keywords:** tardive dyskinesia, BCG vaccine, haloperidol, neurotransmitters, antioxidants

## Abstract

Tardive dyskinesia (TD) is a hyperkinetic movement disorder that displays unusual involuntary movement along with orofacial dysfunction. It is predominantly associated with the long-term use of antipsychotic medications, particularly typical or first-generation antipsychotic drugs such as haloperidol. Oxidative stress, mitochondrial dysfunction, neuroinflammation, and apoptosis are major pathophysiological mechanisms of TD. The BCG vaccine has been reported to suppress inflammation, oxidative stress, and apoptosis and exert neuroprotection via several mechanisms. Our study aimed to confirm the neuroprotective effect of the BCG vaccine against haloperidol-induced TD-like symptoms in rats. The rats were given haloperidol (1 mg/kg, i.p.) for 21 days after 1 h single administration of the BCG vaccine (2 × 10^7^ cfu). Various behavioral parameters for orofacial dyskinesia and locomotor activity were assessed on the 14th and 21st days after haloperidol injection. On the 22nd day, all rats were euthanized, and the striatum was isolated to estimate the biochemical, apoptotic, inflammatory, and neurotransmitter levels. The administration of the BCG vaccine reversed orofacial dyskinesia and improved motor function in regard to haloperidol-induced TD-like symptoms in rats. The BCG vaccine also enhanced the levels of antioxidant enzymes (SOD, GSH) and reduced prooxidants (MDA, nitrite) and pro-apoptotic markers (Cas-3, Cas-6, Cas-9) in rat brains. Besides this, BCG treatment also restored the neurotransmitter (DA, NE, 5-HT) levels and decreased the levels of HVA in the striatum. The study findings suggest that the BCG vaccine has antioxidant, antiapoptotic, and neuromodulatory properties that could be relevant in the management of TD.

## 1. Introduction

Tardive dyskinesia (TD) is a severe neurological condition caused by the prolonged administration of antipsychotic drugs, characterized by impaired orofacial movements, including tongue, lips, and jaws [1,2]. Several studies have documented that typical antipsychotics block the D2 receptor, leading to impaired indirect pathway inhibitory activity. This results in hyperkinetic consequences like facial choreiform movement [3,4]. Prolonged exposure to antipsychotics can lead to the activation of various devastating pathways, like free radical generation, neuroinflammation, and apoptosis, ultimately worsening the patient’s condition [5,6]. The striatum contains GABAergic neurons that help to regulate excitatory activity through direct pathways. However, the long-term intake of antipsychotics can affect the function of GABAergic neurons, leading to the disruption of excitatory and inhibitory signaling in the basal ganglia [1,7]. Preclinical studies indicate that the blocking of the D2 receptor by antipsychotics enhances the release of extracellular dopamine, which facilitates the dopamine metabolism process, leading to the production of dopamine quinones, which can undergo redox cycling. This process can generate reactive oxygen species (ROS) as byproducts [4]. Moreover, ROS is considered the key factor in activating oxidative stress, neuroinflammation, mitochondrial dysfunction, etc., contributing to neuronal cell death [8,9]. These pathophysiological alterations are responsible for initiating involuntary TD-like symptoms in animals. Presently, vesicular monoamine transporter 2 (VMAT-2) inhibitors are the only option used to overcome the release of dopamine from the presynaptic junction and are used for the symptomatic relief of TD [10]. However, treatment with VMAT-2 inhibitors exerts severe side effects, such as depression, parkinsonism, and tiredness [11,12]. This indicates the need to explore newer therapeutic options for TD management.

In 1921, the Bacillus Calmette–Guérin (BCG) vaccine was developed from Mycobacterium bovine species and administered to humans; it protects against Mycobacterium tuberculosis infection [13,14]. The BCG vaccine also has a prolonged immune response, showing a protective effect against breast cancer; therefore, later, it was used as a first-line therapy for non-muscle metastatic bladder cancer [15]. A randomized control trial reported that the BCG vaccine could lower the blood glucose level in type-1 diabetic subjects [16]. Recently, the BCG vaccine has been explored by researchers as a preventive therapy for neurological disorders, and their results indicate that the BCG vaccine has the ability to decrease neurological symptoms and improve motor function in rodents [17,18]. A study revealed that the BCG vaccine increases the immune-mediated production of regulatory T cells (Tregs); these Tregs are known to exert neuroprotective action [19,20]. Studies indicate that vaccination can modify the immune system, reduce inflammation and oxidative stress, and restore the normal functioning of neurons [18,21]. Moreover, it has been reported that the BCG vaccine exerts a neuroprotective effect via the modulation of signaling pathways and attenuates the progression of movement disorders, which include Parkinson’s disease (PD) [22], Alzheimer’s disease (AD) [23], Huntington’s disease (HD) [17], and multiple sclerosis [21]. The evidence indicates that the BCG vaccine may have a neuroprotective effect [17,21].

Previously, researchers developed a haloperidol-induced TD model in rodents, which induced pathological changes such as oxidative stress, mitochondrial dysfunction, apoptosis, neuroinflammation, and neurochemical abnormalities, resulting in the development of TD-like hyperkinetic conditions in animals [7,24]. To date, there is no evidence indicating an effect of the BCG vaccine on TD. For this reason, our research investigated the neuroprotective potential of the BCG vaccine against haloperidol-induced orofacial dyskinesia in rats.

## 2. Materials and Methods

### 2.1. Experimental Animals

A total of 24 male Wistar rats (250–300 g) were housed in polyacrylic cages under suitable laboratory conditions (RT 22 ± 1 °C and RH of 60%) with a 12 h light/dark cycle at the Central Animal House of the Department of Pharmaceutical Sciences and Technology, MRSPTU, Bathinda, Punjab (India). Food and water were provided ad libitum. The protocol followed the guidelines provided by the Institutional Animal Ethics Committee (IAEC), Maharaja Ranjit Singh Punjab Technical University (MRSPTU), and their approval number is MRSPTU/IAEC/2018/001, which was approved on 23/08/2018. During the experimental protocol, care was taken to ensure no variations in treatment by considering age groups among animals.

### 2.2. Drug and Chemicals

The haloperidol (#Cat-H1512-5G) was procured from Sigma-Aldrich, 3050 Spruce Street, Saint Louis, MO 63103, United State America (USA). The BCG vaccine was bought from a registered pharmacy and manufactured by the Serum Institute of India Pvt. Ltd. 212/2, Hadapsar, Off Soli Poonawalla Road, Pune 411028 India. The ELISA kits, such as Caspase-3 (#Cat-K11-0281), Caspase-6 (#Cat-KLR1454), Caspase-9 (#Cat-KLR1898), and IL-6 (#Cat-KB3068), were bought from Krishgen Biosystems, Mumbai, India. All the additional chemicals were obtained from Sisco Research Laboratories Pvt. Ltd. (SRL) (Mumbai, India), Molychem (Mumbai, India), and Sigma-Aldrich USA.

### 2.3. Protocol Schedule

The haloperidol was prepared freshly in normal saline and injected (i.p.) 1 mg/kg daily for three weeks. A single injection of BCG vaccine dissolved in distilled water was administered on the 1st day, 1 h before haloperidol injection. All the behavioral parameters were assessed on the 14th and 21st days. All rats were euthanized on the 22nd day, and the striatum tissue was separated to determine biochemical, cytokine, apoptotic, and neurotransmitter levels. Before experimentation, rats were equally distributed into the following groups, each group consisting of 8 animals, and trained for different behavioral apparatuses (Figure 1).

### 2.4. Experimental Groups

Group 1: Control (normal saline i.p.)

Group 2: Haloperidol (1 mg/kg, i.p.)

Group 3: Haloperidol (1 mg/kg, i.p.) + BCG (2 × 10^7^ cfu, i.p.)

The BCG vaccine group was excluded from the study because previous research had found that BCG vaccine treatment alone had no significant effect compared to a control group [17,25].

### 2.5. Behavioral Assessments

#### 2.5.1. Assessment of Orofacial Movement

A Plexiglass transparent chamber with dimensions of 30 cm × 20 cm × 30 cm was utilized to observe the orofacial movements. Firstly, the animal was allowed to explore the cage for 10 min to reduce the novelty-induced stress. During the trial, animals were placed in the chamber for 10 min. The orofacial movement (VCMs, tongue protrusion, and facial jerking) was assessed by hand-operated counting for the analysis of orofacial dyskinesia, and the counting was paused when the rat started grooming and restarted when it finished [4,7].

#### 2.5.2. Rotarod Activity

The rotarod apparatus manufactured by IMCORP, Ambala, India was utilized to assess the motor coordination of rats; the apparatus contained a rod having a 7 cm diameter and 30 cm length, which was separated into four different sections. Before experimentation, the animals were trained on the apparatus for 180 s with a speed of 25 rpm. Afterward, the trial was conducted, the latency to fall-off time was recorded, and the cutoff time was set at 180 s [26].

#### 2.5.3. Narrow Beam Walk Test (NBW)

The gait abnormalities in the experimental rats were analyzed using a narrow beam apparatus; the apparatus was created using a wooden beam that was 120 cm long, 0.5 mm thick, and 2.0 cm wide, hanging 1 m above the ground. The rats were trained on the narrow beam for 2 min before experimentation. The time that the rats took to cross the beam and the number of foot slips were noted for three trials, and the average was taken as a final value. The cutoff time for the beam cross was set as 60 s [27].

#### 2.5.4. Open Field Test (OFT)

An open field test setup was utilized to analyze locomotor activity; the apparatus was a large wooden box with a 55 cm height and a 90 × 90 cm square bottom, equally divided into 36 different zones. A 2.0 MP camera was fixed on the top of the apparatus and operated using the Maze Master video tracking software UI-version 5.0.0.0. Before experimenting, rats were allowed 10 min to explore the open field. During the trial, the number of crossings, total distance traversed (cm), and active time within 10 min were recorded by the software. This test was carried out once a week for three weeks [26].

### 2.6. Dissection and Homogenization

On the last day of experimentation, all rats were sacrificed by cervical dislocation. The brains were removed and frozen at −80 °C for biochemical investigation. The striatum was separated, and the homogenate was prepared with 10% *w/v* 0.1 M PBS (pH 7.4). The supernatant was obtained after centrifuging at 14,000× *g* for 15 min at 4 °C, and then the supernatant was utilized to analyze antioxidant, inflammatory, and apoptotic markers and neurotransmitters [7].

### 2.7. Biochemical Assessment

#### 2.7.1. Estimation of Malondialdehyde (MDA)

MDA levels were measured in the striatum by following the Wills et al. method. Briefly, an equal quantity of sample homogenate with Tris–HCL buffer was incubated at RT for 2 h; it was then mixed with 10% TCA and centrifuged. The obtained supernatant was separated, TBA was added, and it was then heated in the water bath system for 10 min, and the solution was cooled. The optical density of malondialdehyde (MDA) at 532 nm was measured using a BioTek Microplate reader (BTFLX800TB), Agilent, 5301 Stevens Creek Blvd. Santa Clara, CA 95051, USA [28].

#### 2.7.2. Estimation of Nitrite

The nitrite level as an indication of the formation of nitric oxide was evaluated using Griess reagent with the striatal tissue homogenate. The reading was taken at 540 nm using a BioTek Microplate reader (BTFLX800TB) Agilent, 5301 Stevens Creek Blvd. Santa Clara, CA 95051, USA, and the values were used as μ mol/mg protein [3,22].

#### 2.7.3. Estimation of Reduced Glutathione (GSH)

GSH was measured using Ellman’s method [29]. The glutathione level in the striatum homogenate was determined using a standard curve and displayed as nmol/mg protein.

#### 2.7.4. Estimation of Superoxide Dismutase (SOD)

The SOD concentration was estimated using the procedure of Kono et al., According to the procedure, the nitro blue tetrazolium, EDTA, and sodium carbonate solution were prepared, added with hydroxylamine, and taken as a blank. The sample was added to the same solution, reacting with the hydroxylamine to show the presence of SOD, which was examined every thirty seconds for 2 min at 560 nm using UV spectrophotometers (UV-1800 UV–VIS) Shimadzu Corp. N5-4F, 5F1 Tokudaiji-cho Nishinokyo Nakagyo-ku, Kyoto 604-8445, Japan [30].

#### 2.7.5. Estimation of Catalase

The catalase activity was measured using a method developed by Luck et al. It involves the breakdown of H_2_O_2_ in the presence of the catalase enzyme, which was performed at 240 nm using a UV spectrophotometer (UV-1800 UV–VIS) Shimadzu Corp. N5-4F, 5F1 Tokudaiji-cho Nishinokyo Nakagyo-ku, Kyoto 604-8445, Japan [31]. The readings were expressed as μ mol H_2_O_2_ decomposed/min/mg protein.

#### 2.7.6. Protein Estimation

The Lowry method was utilized for protein determination [32].

#### 2.7.7. Estimation of IL-6, Cas-3, Cas-6, and Cas-9 Levels

The apoptotic markers, such as Caspase-3, Caspase-6, Caspase-9, and inflammatory cytokine IL-6, were assessed using sandwich ELISA kits, following the instructions provided by Krishgen Biosystem. The quantification was performed using an iMark ELISA reader (Model No: 21350), and the sample values were calculated using the standard curve [26].

### 2.8. Neurochemical Analysis

The neurotransmitters (dopamine (DA), serotonin (5-HT), and norepinephrine (NE)) and their metabolite, homovanillic acid (HVA), were measured using a Waters HPLC-ECD (model no: 2465) instrument. All neurochemicals were determined by applying the method described by Datta et al. [7,24].

### 2.9. Statistical Analysis

The data were analyzed as the mean ± S.D using the GraphPad Prism software, version 8.0.1. The behavioral activity was assessed using two-way ANOVA followed by the Tukey Post-hoc test for multiple comparisons. One-way ANOVA was used for biochemical, neurochemical, inflammatory, and apoptotic marker determination, followed by Dunnett’s test. *p* < 0.05 was considered statistically significant.

## 3. Results

### 3.1. Effect of BCG Vaccine on VCMs, TPs, and FJs in Haloperidol-Induced TD in Rats

Haloperidol administration significantly (*p* < 0.01) increased the orofacial dyskinesia (VCMs, FJMs, TPs) compared with the control group. The BCG vaccine (2 × 10^7^ cfu, i.p.) pretreatment significantly (*p* < 0.01) decreased VCMs (Figure 2A), TPs (Figure 2B), and FJs (Figure 2C) on day 14 and 21 as compared to the haloperidol-alone group.

### 3.2. Effect of BCG Vaccine on Rotarod and NBW Activity in Haloperidol-Induced TD Rats

Haloperidol administration significantly (*p* < 0.01) reduced the fall time and enhanced the time to cross the beam and foot slips compared to the control group. Nevertheless, after administering the BCG vaccine (2 × 10 ^7^cfu, i.p.), there was a significant (*p* < 0.01) increase in fall-off time (Figure 3A) and a decrease in the time to cross the beam (Figure 3B) and the number of foot slips (Figure 3C) on day 14 and 21 as compared with the haloperidol-alone group.

### 3.3. Effect of BCG Vaccine on Locomotor Activity in Haloperidol-Induced TD Rats

Haloperidol administration showed a substantial (*p* < 0.01) decrease in motor activity in the OFT compared to the control group. Moreover, the BCG vaccine (2 × 10^7^ cfu, i.p.) treatment showed a considerable (*p* < 0.01) rise in the number of crossings (Figure 4A), total distance-traveled (Figure 4B), and active time (Figure 4C) in motor activity (Figure 5) on day 14 and 21 as compared with the haloperidol-alone group.

### 3.4. Effect of BCG Vaccine on Oxidative Stress Markers in Haloperidol-Induced TD Rats

Haloperidol administration in rats significantly (*p* < 0.01) enhanced the MDA and nitrite levels in the striatal tissues compared to the control group. Pre-treatment with the BCG vaccine (2 × 10^7^ cfu, i.p.) substantially (*p* < 0.01) decreased the MDA (Figure 6A) and nitrite (Figure 6B) levels compared to the haloperidol-alone group.

### 3.5. Effect of BCG Vaccine on Antioxidants (GSH, SOD, and Catalase) in Haloperidol-Induced TD Rats

Haloperidol-injected rats showed a remarkable (*p* < 0.01) reduction in GSH levels, SOD activity, and catalase units in the striatum compared to the control group. However, the BCG vaccine (2 × 10^7^ cfu, i.p.) significantly (*p* < 0.01) restored the GSH (Figure 7A), SOD activity (Figure 7B), and catalase (Figure 7C) units in comparison with the haloperidol-alone group.

### 3.6. Effect of BCG Vaccine on Apoptotic Markers (Cas-3, Cas-6, and Cas-9) and Inflammatory Cytokines (IL-6) in Haloperidol-Induced TD Rats

Haloperidol-treated rats showed a substantial (*p* < 0.01) increase in IL-6, Cas-3, Cas-6, and Cas-9 in the striatum compared to the control group. However, in animals pre-treated with the BCG vaccine (2 × 10^7^ cfu, i.p.) along with haloperidol, the treatment significantly (*p* < 0.01) reversed the increase in Cas-3 (Figure 8A), Cas-6 (Figure 8B), Cas-9 (Figure 8C), and IL-6 (Figure 8D) levels in comparison to the haloperidol-alone group.

### 3.7. Effect of BCG Vaccine on Haloperidol-Induced TD Alterations in Neurotransmitters

Haloperidol-injected rats showed a significant (*p* < 0.01) decrease in DA, NE, and 5-HT levels and increased metabolite HVA in the striatum compared to the control group. The BCG vaccine (2 × 10^7^ cfu, i.p.) given alongside haloperidol substantially (*p* < 0.01) increased the levels of DA (Figure 9A), NE (Figure 9B), and 5-HT (Figure 9C) and decrease the HVA (Figure 9D) as compared to the haloperidol-alone group.

## 4. Discussion

This study investigated the neuroprotective property of the BCG vaccine against haloperidol-induced TD-like orofacial dyskinesia and motor abnormalities. Haloperidol is a first-generation typical antipsychotic agent used to manage schizophrenia [33]. However, its long-term use is reported to cause hyperkinetic orofacial dyskinesia, also known as TD [34]. Haloperidol can persistently block dopaminergic D2 receptors and increase their sensitivity towards dopamine. This alteration amplifies the extracellular dopamine concentration along with its metabolism, leading to the generation of free radicals, which are a contributing factor for oxidative stress in the striatum [35]. An earlier preclinical study reported that haloperidol’s excessive oxidative damage can disrupt the striatal neurons’ function, resulting in TD-like symptoms such as VCMs, TP, and FJ in experimental rats [6]. Furthermore, haloperidol exposure impaired rodents’ locomotor activity, motor coordination, and gait balance [36].

Therefore, researchers have developed a haloperidol-induced rodent model that provides similar symptomatic conditions to TD [37].

This study used a haloperidol-induced TD-like animal model to investigate the neuroprotective role of the BCG vaccine. Our findings indicated that the BCG vaccine reversed haloperidol-induced toxicity, evident in the experimental rats’ reduction in VCMs, TP, and FJ behavior. Moreover, the BCG vaccine improved locomotor activity, motor coordination, and neuromuscular function in haloperidol-treated animals. Similarly, an earlier study observed that the BCG vaccine could improve neurotoxin-mediated locomotor dysfunction in rodents [38].

Preclinical studies have reported that haloperidol treatment increases free radicals’ formation and raises the oxidative burden, subsequently leading to mitochondrial dysfunction [39,40]. Some evidence supports the notion that haloperidol increases oxidative stress markers and reduces the antioxidant enzymatic activity in the rodent’s brain [17,36]. Another study demonstrated that the BCG vaccine had the potential to decrease MDA and nitrite concentrations while boosting antioxidants like GSH and SOD in 3-nitro propionic acid (3-NPA)-induced HD rats [17]. The findings of Senousy et al. 2022 [17] identified that the BCG vaccine inhibited the PI3k/Akt/mTOR signaling pathway and was involved in the improvement of mitochondrial function and restored the antioxidant levels in the 3-NPA-treated rat brain, which shows the neuroprotective potential of the BCG vaccine. Similarly, in our previous study, BCG reduced oxidative stress markers and improved locomotor activity in QA-induced HD rats [37]. Our study findings also indicate that the BCG vaccine has an antioxidant property, as it restored antioxidants (GSH, SOD, and catalase) and lowered oxido-nitrosative stress markers (MDA and nitrite) in the rat striatum, providing neuroprotection from oxidative stress generated by haloperidol.

Previously, it was found that haloperidol treatment increased the IL-6 in the striatum, which upregulated the neuroinflammatory cascades in the nigrostriatal pathway. Researchers have reported the non-specific anti-inflammatory effect of the BCG vaccine against autoimmune disorders by limiting microglia activation and providing a neuroprotective effect in the experimental autoimmune encephalomyelitis (EAE) mice model [21]. Another study reported that BCG vaccine injection downregulated the expression of IL-6 in a Japanese encephalitis mice model [41]. Furthermore, our results suggest that three weeks of haloperidol treatment enhanced proinflammatory markers (IL-6), whereas BCG vaccination reduced the IL-6 in the striatum, thus limiting haloperidol-induced neuroinflammation in rodents. Our previous studies found that animals treated with haloperidol displayed a significant increase in apoptotic markers, including caspase-3, -6, and -9, responsible for striatal dopaminergic neuronal loss in experimental animals [6,7,24]. Similarly, evidence from a clinical study showed that antipsychotic treatment enhanced the release of apoptotic markers in TD patients’ brains [42,43]. We identified that BCG vaccination mitigated haloperidol-associated increases in caspase 3-, -6, and -9 in the animals and reduced the TD-like symptoms.

Neurotransmitters are an important aspect of movement disorders. They are involved in the initiation/control of movement and have a strong correlation with TD. Therefore, in earlier studies, researchers reported that haloperidol treatment impaired rodents’ neurotransmitters [7,35]. Earlier, it was found that the administration of haloperidol altered the neurotransmitter levels in the brain, specifically DA, 5-HT, NE, and their metabolites, which are essential markers for hyperkinetic movement disorder [44]. Previously, Lacan et al. 2013 [22] repurposed the BCG vaccine in an MPTP-induced PD model, and their results indicated that treatment with the BCG vaccine restored neurotransmitter levels to prevent dopaminergic neuronal degeneration in mice injected with MPTP. Our results indicate that haloperidol-induced TD rats had reduced levels of neurotransmitters (DA, 5-HT, NE) with subsequent increases in their metabolites (HVA), signifying the rapid degradation of these neurotransmitters. The findings support that the BCG vaccine can potentially restore neurotransmitters and reduce their metabolites in rats treated with haloperidol. In summary, our findings indicate that the BCG vaccine has antioxidant, anti-inflammatory, and anti-apoptotic properties, as well as restoring the neurotransmitter levels to ameliorate TD-like symptoms in animals (Figure 10).

Previously, Gofrit et al., in their epidemiology survey, concluded that people living in countries without BCG vaccination policies have a higher risk of developing neurological disorders than people living in countries with BCG vaccination policies. However, it is possible that the BCG vaccine also reduces the risk of developing TD [18,45]. Further research is required to understand the interaction between antipsychotics and BCG vaccination, as well as the protective mechanism of BCG vaccination.

## 5. Conclusions

The study suggests that the BCG vaccine can improve motor coordination and reduce ROS generation. Moreover, the BCG vaccine has antiapoptotic and anti-inflammatory properties. The study suggests that the BCG vaccine could help to manage TD and other neurological disorders. Further study is, however, required to explore the molecular mechanism of BCG-mediated neuroprotection.

## Figures and Tables

**Figure 1 biomolecules-13-01667-f001:**
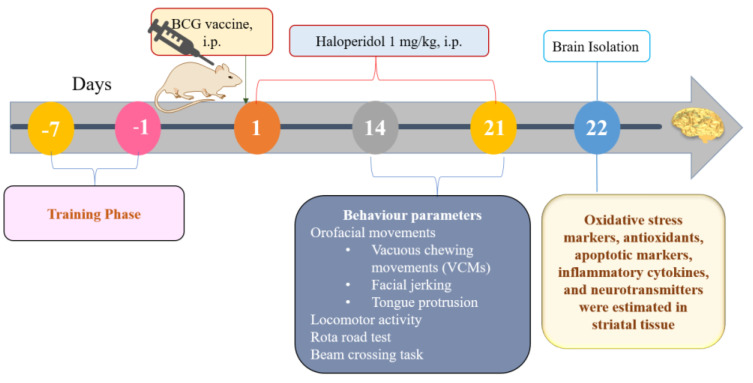
Experimental schedule.

**Figure 2 biomolecules-13-01667-f002:**
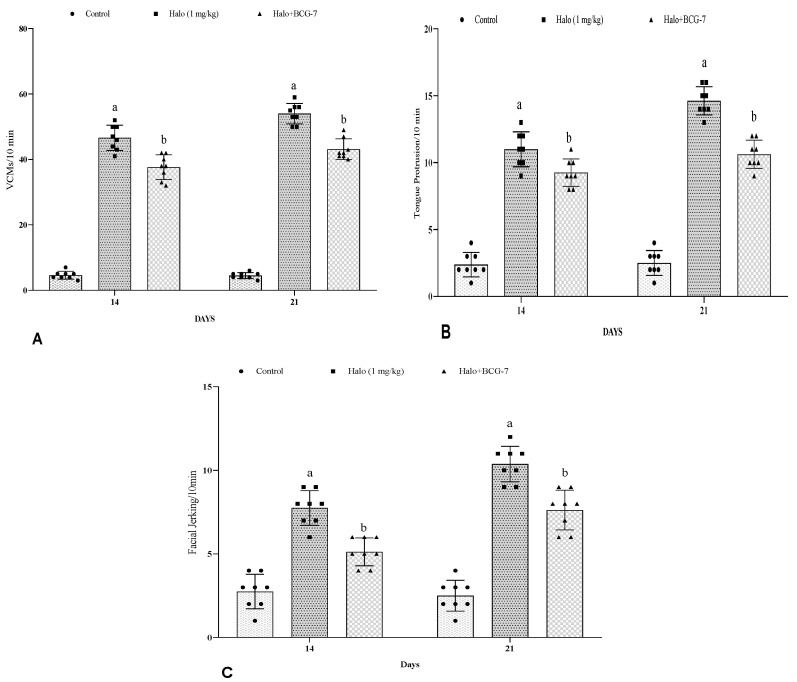
Effect of BCG vaccine on VCMs (**A**), TPs (**B**), FJs (**C**), in haloperidol-induced TD rats. Data are expressed as mean ± S.D analyzed by two-way ANOVA followed by Tukey’s post hoc test, ap < 0.0001 Control vs. Halo, bp < 0.01 Halo vs. BCG-7. a denotes comparison to Control and b denotes comparison to Halo group. (Halo: haloperidol; BCG: Bacillus Calmette–Guérin; VCMs: vacuole chewing movements; TD: tardive dyskinesia; TPs: tongue protrusion; FJs: facial jerking movement; S.D: standard deviation; ANOVA: analysis of variance).

**Figure 3 biomolecules-13-01667-f003:**
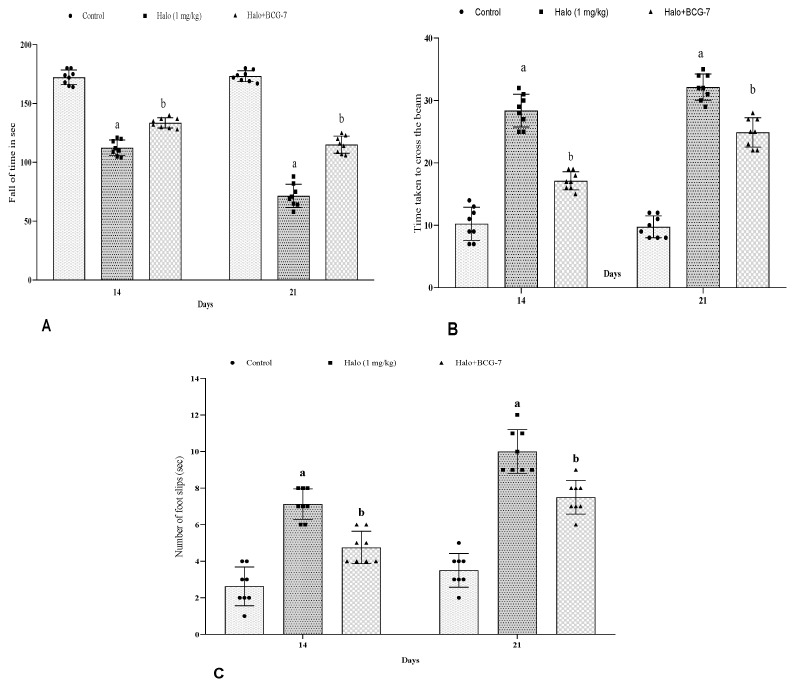
Effect of BCG vaccine on fall-off time (**A**), transfer latency (**B**), and foot slips (**C**) in haloperidol-induced TD rats. Data are expressed as mean ± S.D analyzed by two-way ANOVA followed by Tukey’s test, ap < 0.0001 Control vs. Halo, bp < 0.01 Halo vs. BCG-7. a denotes comparison to Control and b denotes comparison to Halo group. (Halo: haloperidol; BCG: Bacillus Calmette–Guérin; S.D: standard deviation; ANOVA: analysis of variance).

**Figure 4 biomolecules-13-01667-f004:**
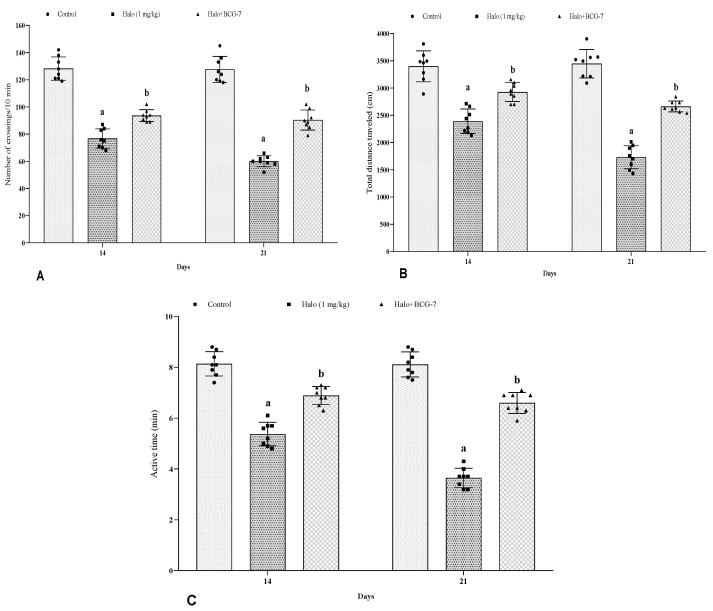
Effect of BCG vaccine on locomotor activity, namely number of crossings (**A**), total distance traveled (**B**), and active time (**C**), in haloperidol-induced TD rats. Data are expressed as mean ± S.D analyzed by two-way ANOVA followed by Tukey’s post hoc test, ap < 0.01 Control vs. Halo, bp < 0.01 Halo vs. BCG-7. a denotes comparison to Control and b denotes comparison to Halo group. (Halo: haloperidol; BCG: Bacillus Calmette–Guérin; S.D: standard deviation; ANOVA: analysis of variance).

**Figure 5 biomolecules-13-01667-f005:**
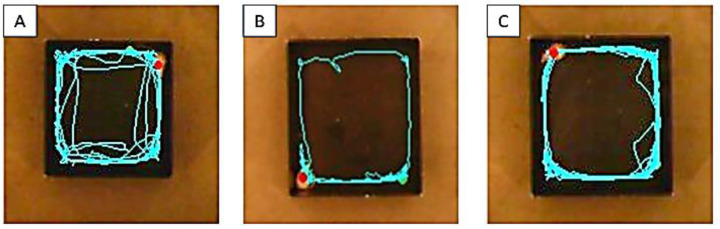
Trajectory images of the open field test (OFT): (**A**) Control, (**B**) Halo 1 mg/kg, (**C**) Halo + BCG-7.

**Figure 6 biomolecules-13-01667-f006:**
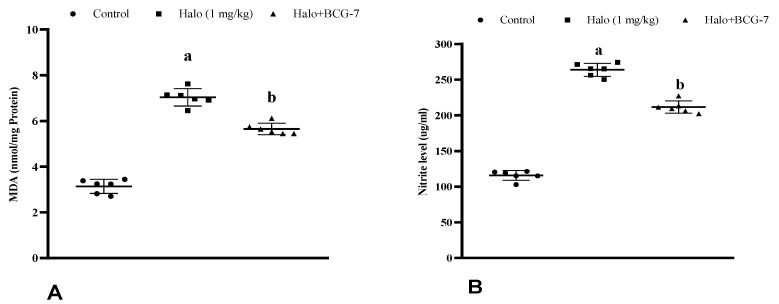
Effect of BCG vaccine on oxidative stress markers, namely MDA (**A**) and nitrite (**B**), in haloperidol-induced TD rats. Data are expressed as mean ± S.D analyzed by one-way ANOVA followed by Dunnett’s post hoc test, ap < 0.01 Control vs. Halo, bp < 0.01 Halo vs. BCG-7. a denotes comparison to Control and b denotes comparison to Halo group. (Halo: haloperidol; BCG: Bacillus Calmette–Guérin; MDA: malondialdehyde; S.D: standard deviation; ANOVA: analysis of variance).

**Figure 7 biomolecules-13-01667-f007:**
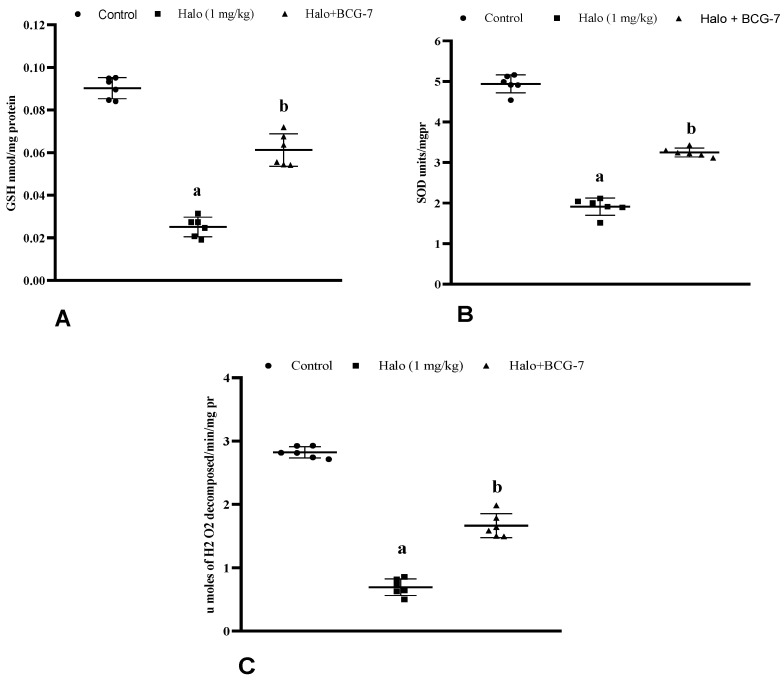
Effect of BCG vaccine on antioxidant enzymes GSH (**A**), SOD (**B**), and catalase (**C**) in haloperidol-induced TD rats. Data are expressed as mean ± S.D analyzed by one-way ANOVA followed by Dunnett’s post hoc test, ap < 0.0001 Control vs. Halo, bp < 0.01 Halo vs. BCG-7. a denotes comparison to Control and b denotes comparison to Halo group. (Halo: haloperidol; BCG: Bacillus Calmette–Guérin; GSH: glutathione; SOD: superoxide dismutase; S.D: standard deviation; ANOVA: analysis of variance).

**Figure 8 biomolecules-13-01667-f008:**
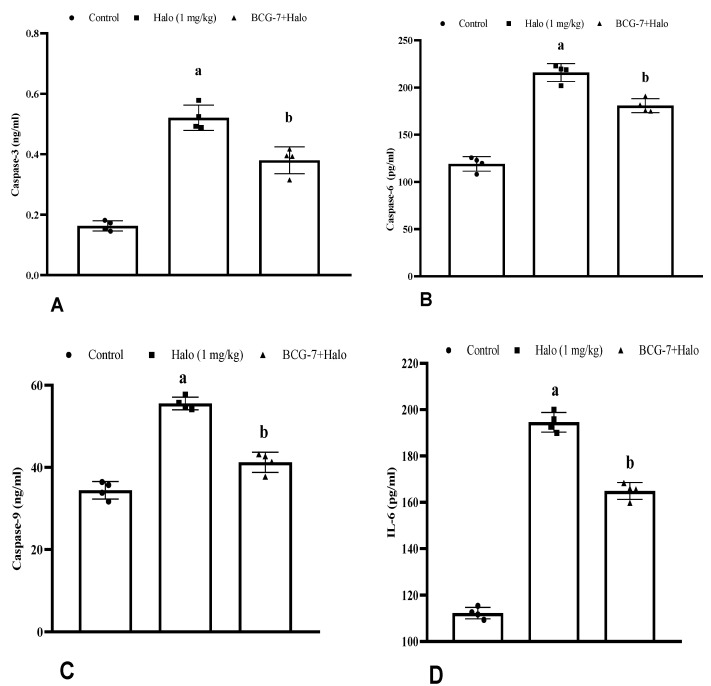
Effect of BCG vaccine on apoptotic markers Caspase-3 (**A**), Caspase-6 (**B**), Caspase-9 (**C**), and (**D**) IL-6 in haloperidol-induced TD rats. Data are expressed as mean ± S.D analyzed by one-way ANOVA followed by Dunnett’s post hoc test, ap < 0.01 Control vs. Halo, ap < 0.01 Halo vs. BCG-7. a denotes comparison to Control and b denotes comparison to Halo group. (Halo: haloperidol; BCG: Bacillus Calmette–Guérin; Cas: caspase; IL-6: interleukin-6; S.D: standard deviation; ANOVA: analysis of variance).

**Figure 9 biomolecules-13-01667-f009:**
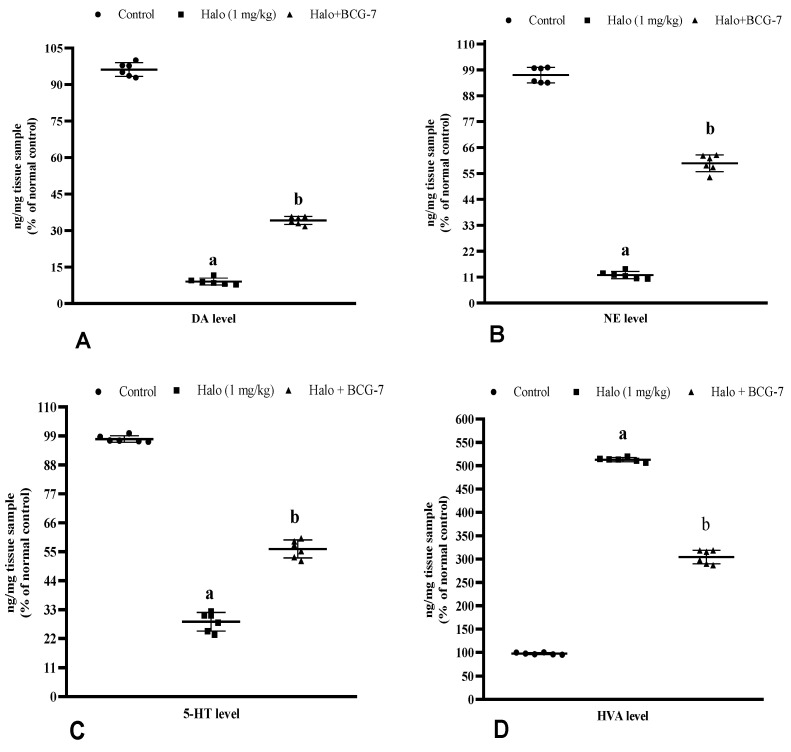
Effect of BCG vaccine on neurotransmitters DA (**A**), NE (**B**), 5-HT (**C**), and its metabolite HVA (**D**) in haloperidol-induced TD rats. Data are expressed as mean ± S.D analyzed by one-way ANOVA followed by Dunnett’s post hoc test, ap < 0.01 Control vs. Halo, bp < 0.01 Halo vs. BCG-7. a denotes comparison to Control and b denotes comparison to Halo group. (Halo: haloperidol; BCG: Bacillus Calmette–Guérin; DA: dopamine; NE: norepinephrine; 5-HT: serotonin; HVA: homovanillic acid; S.D: standard deviation; ANOVA: analysis of variance).

**Figure 10 biomolecules-13-01667-f010:**
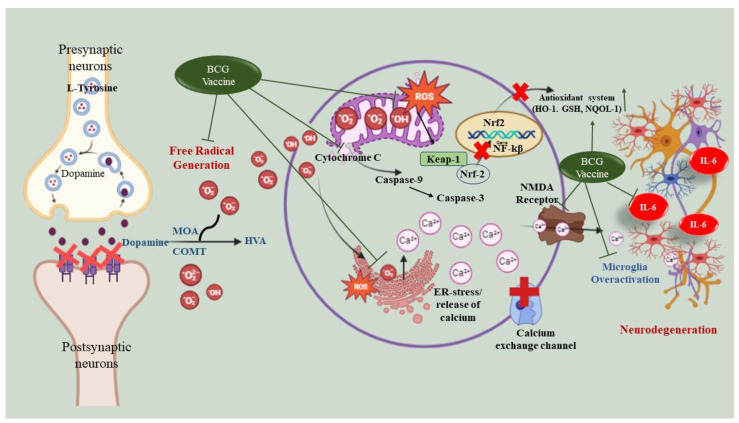
Proposed mechanism of BCG vaccine against haloperidol-induced ROS generation and pathological changes.

## Data Availability

All data generated or analyzed during this study are included in this article.

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
