# Peer review of "Bacillus Calmette–Guérin Vaccine Attenuates Haloperidol-Induced TD-like Behavioral and Neurochemical Alteration in Experimental Rats"

_biomolecules, 2023, doi:10.3390/biom13111667_

Round 1

Reviewer 1 Report

Comments and Suggestions for Authors

In this research, Yedke et al. undertook an investigation aimed at substantiating the neuroprotective efficacy of the BCG vaccine against haloperidol-induced tardive dyskinesia (TD)-like symptoms in rats. Collectively, the findings expound that the administration of the BCG vaccine effectively mitigated orofacial dyskinesia, ameliorated motor function in the context of haloperidol-induced TD-like symptoms in rats, elevated the activity of antioxidant enzymes, attenuated pro-oxidant elements, reduced pro-apoptotic markers within the rat brain, reinstated neurotransmitter levels, and diminished HVA levels in the striatum. Based on the data thus unveiled, the authors posit that the BCG vaccine possesses antioxidant, antiapoptotic, and neuromodulatory attributes that could hold significance in the management of TD. In view of these discoveries, I consider the subject matter of this investigation and the presented dataset to be compelling. However, it is essential to address certain specific concerns before granting formal acceptance of the manuscript. Below, I provide detailed comments pertaining to the content presented within the manuscript.

1. I have reservations regarding the statistical analysis employed, specifically the utilization of a two-way ANOVA. Could you please specify the independent variables under consideration? I kindly request that this aspect be clarified in the forthcoming revision.

2. It is imperative that the authors include the results of normality and homogeneity of variance tests in their reporting.

3. The results presented in this study are indeed intriguing. However, it is essential to provide mechanistic data to elucidate the mode of action of BCG. The current version appears to be predominantly phenomenological in nature, and it would greatly benefit from the inclusion of mechanistic and molecular evidence (new data; new experiments).

4. The discussion section requires enhancement to delve tentatively into potential mechanisms underlying the protective effects of BCG. The current discussion predominantly revolves around comparisons with other scientific reports, both with regard to the model used and the protective efficacy of BCG in various protocols.

5. There is suggestive evidence to support the hypothesis that individuals who have received the BCG vaccine exhibit a reduced probability of developing TD-like symptoms. I recommend that this be taken into account during the revision process.

Author Response

Point-to-point response to reviewer comments (Biomolecules-2644348)

Reviewer #1

  1. I have reservations regarding the statistical analysis employed, specifically the utilization of a two-way ANOVA. Could you please specify the independent variables under consideration? I kindly request that this aspect be clarified in the forthcoming revision

Response: The two-way ANOVA was used to analyse behavioural parameters which includes two independent variables (days and treatment).

  1. It is imperative that the authors include the results of normality and homogeneity of variance tests in their reporting

Response: Shapiro-Wilk test was used to assess the normality distribution and the data passed the normality test.

  1. The results presented in this study are indeed intriguing. However, it is essential to provide mechanistic data to elucidate the mode of action of BCG. The current version appears to be predominantly phenomenological in nature, and it would greatly benefit from the inclusion of mechanistic and molecular evidence (new data; new experiments).

Response: The current study is part of a research project which was completed. In future research, we will explore the molecular mechanism of BCG vaccines.

  1. The discussion section requires enhancement to delve tentatively into potential mechanisms underlying the protective effects of BCG. The current discussion predominantly revolves around comparisons with other scientific reports, both with regard to the model used and the protective efficacy of BCG in various protocols.

Response: Additional information regarding the protective effect of BCG against on other models has been written in the discussion section.  

  1. There is suggestive evidence to support the hypothesis that individuals who have received the BCG vaccine exhibit a reduced probability of developing TD-like symptoms. I recommend that this be taken into account during the revision process.

Response: Recommended point has been discussed in discussion section

Reviewer 2 Report

Comments and Suggestions for Authors

In this manuscript, Yedke et al. have evaluated the protective action of Bacillus Calmette-Guérin vaccine (BCG) against tardive dyskinesia (TD) induced by the antipsychotic haloperidol. For this, they have used behavioural and biochemical determinations to evaluate motor activity along to oxidative stress and apoptosis parameters and monoamine content. According to the authors' bibliographical production (see ref 18 and 37), this paper is part of a global project aimed at evaluating the protective activity of BCG vaccine on various neurological conditions. Broadly speaking the manuscript is well-written and the methods adequately described.

Here are my comments:

1-The major comment arises from the fact that the per se effect of BCG vaccine is not shown here on all the parameters tested. Is the injection of BCG vaccine having an effect by itself on the parameters tested ? Indeed, it is difficult to evaluate whether BCG vaccine is effectively reversing the action of haloperidol and/or whether it modifies by itself the behavioural and biochemical parameters. Therefore the experimental group BCG vaccine is missing. I recommend that the authors test at least the motor activity on a group of rats treated only with BCG vaccine.

2-The reversing action of BCG vaccine is partial, ranging from high to modest according the parameter tested. The authors should comment on these discrepancies.

3-With regards to the recapitulative scheme shown in fig 10, the Nrf2 pathway is shown. In fact there are no data relative to this pathway in the present study. How is the BCG vaccine effectively acting on it to induce an antioxydant effect ?  

Author Response

Reviewer # 2

1 The major comment arises from the fact that the per se effect of BCG vaccine is not shown here on all the parameters tested. Is the injection of BCG vaccine having an effect by itself on the parameters tested? Indeed, it is difficult to evaluate whether BCG vaccine is effectively reversing the action of haloperidol and/or whether it modifies by itself the behavioural and biochemical parameters. Therefore, the experimental group BCG vaccine is missing. I recommend that the authors test at least the motor activity on a group of rats treated only with BCG vaccine.

Response: Previously studies did not observe any significant effect of BCG vaccine per se group compared to control group. Thus, according to 3R principals, we exclude perse group in this study.

  • Yang, J., Qi, F., & Yao, Z. (2016). Neonatal Bacillus Calmette-Guérin vaccination alleviates lipopolysaccharide-induced neurobehavioral impairments and neuroinflammation in adult mice. Molecular Medicine Reports, 14(2), 1574-1586.
  • Senousy, M. A., Hanafy, M. E., Shehata, N., & Rizk, S. M. (2022). Erythropoietin and Bacillus Calmette–Guérin Vaccination Mitigate 3-Nitropropionic Acid-Induced Huntington-like Disease in Rats by Modulating the PI3K/Akt/mTOR/P70S6K Pathway and Enhancing the Autophagy. ACS Chemical Neuroscience, 13(6), 721-732.

2 The reversing action of BCG vaccine is partial, ranging from high to modest according the parameter tested. The authors should comment on these discrepancies.

Response: We have addressed this point in discussion

3 With regards to the recapitulative scheme shown in fig 10, the Nrf2 pathway is shown. In fact, there are no data relative to this pathway in the present study. How is the BCG vaccine effectively acting on it to induce an antioxidant effect?

Response: Figure 10 is an observation based hypothetical mechanism, which proposed that BCG vaccination could exert antioxidant effect via Nrf2 mediated signaling pathways and enhance the antioxidant levels that were observed in the study. In our future study, we will investigate the molecular mechanism of BCG vaccination.

Reviewer 3 Report

Comments and Suggestions for Authors

Reviewer comments

Manuscript ID.:  biomolecules-2644348
Title:     Bacillus Calmette-Guérin vaccine attenuates haloperidol-induced TD-like behavioral and neurochemical alteration in experimental rats International Journal of Molecular Sciences

The review article is informative. Authors needs to address the minor comments before acceptance are as follows;

Line 24; Change the word slaughtered to euthanised or sacrificed.

Line 85; Justification required for conducting experiment only with male Wistar rats.

Line 90; IAEC held institution name needs to mention.

Line 92; How no variation among the groups was confirmed/ judged using what method?

Line 100; During trial phase or on day 1 pretreatment with Haloperidol, the behavioural parameters need to assess, if assessed good to present with data.

Line 101-102; Kindly added scientific justification in the result section with BCG ADME data and Haloperidol-BCG cross reactivity result. As both injections given in IP will give support for robust study design.

Line 104; Change the word slaughtered to euthanised or sacrificed.

Line 106; Any methodological approach followed for equal distribution of animal groups required to specify.

Line 109; Figure 1, the training phase should mention as -7 to -1 in the diagrammatic representation.

Line 131; Requires to mention the name, specification and detail of instrument used for the Narrow beam walk experiment.

Line 138; Requires to mention the name, specification and detail of instrument used for the OPT experiment.

Line 161; Mention the methodological reference for Griess reagent and nitric oxide assay.

Line 174; Luck et al, in the reference section is not available, the reference needs to check fully with proper correlation.

Line 183; The sentence is incomplete.

Instruments name, make and model needs to mention for all instruments used for the quantification.

Line 188; The reference didn't describe any methodological analysis for neurotransmitters, check the reference cited for this analysis.

Line 190; Mention the statistical software used with version details.

Line 203; Remove space (S. D) for the standard deviation needs to maintain the uniformity in other sections also.

 Line 208; Elaborate abbreviation, if conducted neurobehavioral analysis needs to mention in the materials and methods.

Line 214; Fig 3B Image text (days) alignment needs to fix.

Line 246; Reference no (7) is mentioned here is valid or irrelevant, check once.

Line 290; In discussion section, the futuristic approach of the research needs to be discussed in the section and suggestion for the BCG vaccine.

Comments on the Quality of English Language

Can improve.

Author Response

Reviewer # 3

Line 24; Change the word slaughtered to euthanised or sacrificed.

Response: slaughtered has been changed to euthanised

Line 85; Justification required for conducting experiment only with male Wistar rats.

Response: The females have higher levels of estrogen that exerts neuroprotective effect. Therefore, to avoid biasness, we have used only male Wistar rats

Line 90; IAEC held institution name needs to mention.

Response: Institution name has been added

Line 92; How no variation among the groups was confirmed/ judged using what method?

Response:  The Shapiro-Wilk test was used to confirm the normality and homogeneity.

Line 100; During trial phase or on day 1 pretreatment with Haloperidol, the behavioural parameters need to assess, if assessed good to present with data.

Response: During the trial animals were trained for apparatus, we didn’t administer any drug during trial. On the first day of treatment, we did not observe any significant changes in the animals.

Line 101-102; Kindly added scientific justification in the result section with BCG ADME data and Haloperidol-BCG cross reactivity result. As both injections given in IP will give support for robust study design.

Response: The point is valid but there is no clinical data on haloperidol with BCG vaccine. The cross reactivity between haloperidol with BCG vaccine is a part of further research. Furthermore, we have discussed some relevant studies in the discussion section.

Line 104; Change the word slaughtered to euthanised or sacrificed.

Response: slaughtered has been changed to euthanised

Line 106; Any methodological approach followed for equal distribution of animal groups required to specify.

Response: Randomization method was used for equal distribution of animals

Line 109; Figure 1, the training phase should mention as -7 to -1 in the diagrammatic representation.

Response: In Figure 1, changes have been made

Line 131; Requires to mention the name, specification and detail of instrument used for the Narrow beam walk experiment.

Response: The Narrow Beam walk was prepared with wooden material by local manufacturer and the dimension has been mentioned in the method section.

Line 138; Requires to mention the name, specification and detail of instrument used for the OPT experiment.

Response: Information has been provided

Line 161; Mention the methodological reference for Griess reagent and nitric oxide assay.

Response: Reference cited

Line 174; Luck et al, in the reference section is not available, the reference needs to check fully with proper correlation.

Response: Luck et al., original reference has been cited

Line 183; The sentence is incomplete. Instruments name, make and model needs to mention for all instruments used for the quantification.

Response: Instrument details has been provided

Line 188; The reference didn't describe any methodological analysis for neurotransmitters, check the reference cited for this analysis.

Response: The reference has been rechecked and corrected

Line 190; Mention the statistical software used with version details.

Response: Information provided

Line 203; Remove space (S. D) for the standard deviation needs to maintain the uniformity in other sections also.

Response: Corrected

Line 208; Elaborate abbreviation, if conducted neurobehavioral analysis needs to mention in the materials and methods.

Response: Correction has been done

Line 214; Fig 3B Image text (days) alignment needs to fix.

Response: Corrected

Line 246; Reference no (7) is mentioned here is valid or irrelevant, check once.

Response: check and corrected

Line 290; In discussion section, the futuristic approach of the research needs to be discussed in the section and suggestion for the BCG vaccine.

Response: Futuristic approach has been added in discussion section

Round 2

Reviewer 2 Report

Comments and Suggestions for Authors

In their reply to reviewers comments, the authors have partially answered to my queries. With regards to my major comment, i.e. the absence of a real BCG vaccine per se control in their experiments, nevertheless, they reply by referencing relevant papers on the BCG vaccine effects. I recommend that they introduce a statement referencing to these two papers in the Materials section in order to explain why they did not include an experimental group investigating the per se effects of BCG vaccine (experimental group design  paragraph 2.4, line 129).

Author Response

# Reviewer 2

  1. In their reply to reviewers comments, the authors have partially answered to my queries. With regards to my major comment, i.e. the absence of a real BCG vaccine per se control in their experiments, nevertheless, they reply by referencing relevant papers on the BCG vaccine effects. I recommend that they introduce a statement referencing to these two papers in the Materials section in order to explain why they did not include an experimental group investigating the per se effects of BCG vaccine (experimental group design  paragraph 2.4, line 129).

Response: A statement for exclusion of BCG vaccine perse group has been written and relevant reference has been cited in experiment group paragraph 2.4, line 120-122
